# Structural insight into the human SID1 transmembrane family member 2 reveals its lipid hydrolytic activity

Dandan Qian[1,6], Ye Cong[2,3,4,5,6], Runhao Wang[1], Quan Chen [1]✉, Chuangye Yan[2,3,4,5]✉ & Deshun Gong [1]✉

The systemic RNAi-defective (SID) transmembrane family member 2 (SIDT2) is a putative nucleic acid channel or transporter that plays essential roles in nucleic acid transport and lipid metabolism. Here, we report the cryo-electron microscopy (EM) structures of human SIDT2, which forms a tightly packed dimer with extensive interactions mediated by two previously uncharacterized extracellular/luminal β-strand-rich domains and the unique transmembrane domain (TMD). The TMD of each SIDT2 protomer contains eleven transmembrane helices (TMs), and no discernible nucleic acid conduction pathway has been identified within the TMD, suggesting that it may act as a transporter. Intriguingly, TM3-6 and TM9-11 form a large cavity with a putative catalytic zinc atom coordinated by three conserved histidine residues and one aspartate residue lying approximately 6 Å from the extracellular/luminal surface of the membrane. Notably, SIDT2 can hydrolyze C18 ceramide into sphingosine and fatty acid with a slow rate. The information presented advances the understanding of the structure-function relationships in the SID1 family proteins.

RNA interference (RNAi) is a Nobel prize-winning technique that allows researchers to study the molecular mechanisms underlying fundamental biological processes. A remarkable property of RNAi in *C. elegans* is its ability to spread the silencing RNA from an initial site to adjacent tissues and even cell progeny, a phenomenon known as systemic RNAi[1], allowing the initiation of RNAi by soaking nematodes with double-stranded RNA (dsRNA)[2] or by cultivating worms on bacteria expressing dsRNA[3]. Craig P. Hunter and coworkers found that systemic RNAi is dependent on a member of the systemic RNAi-defective (SID) family, SID1, which functions as a channel that passively transports dsRNA into cells without consumption of ATP[4,5].

A range of organisms that lack systemic RNAi, including mammals, but exhibit striking SID gene conservation[4,6], raising the question, do human orthologs have nucleic acid transport ability similar to that of *C. elegans*. Two orthologs of SID1, SIDT1 and SIDT2, are found in mammals. Human SIDT1 can facilitate small interfering RNA (siRNA) uptake and enhance gene silencing efficacy in human systems[7]. SIDT1 is also required for cellular uptake of cholesterol-conjugated siRNAs[8]. MicroRNA-21 (miRNA21), a well-characterized "oncogenic" miRNA, is widely overexpressed in human cancer and promotes therapeutic resistance in a number of human cancers[9]. It has been reported that SIDT1-mediated intercellular transfer of miRNA21 is a driver of resistance to the nucleoside analog gemcitabine in human adenocarcinoma cells[10]. Strikingly, SIDT1 was recently found to mediate dietary miRNA absorption in the mammalian stomach[11]. Intriguingly, SIDT2, which predominantly localizes to lysosome and also localizes in part to endolysosome, can transport extracellular dsRNA into the cytoplasm for innate immune recognition. SIDT2-deficient mice exposed to

[1]State Key Laboratory of Medicinal Chemical Biology and College of Life Sciences, Nankai University, Tianjin 300350, China. [2]School of Life Sciences, Tsinghua University, Beijing 100084, China. [3]Tsinghua-Peking Joint Center for Life Sciences, Tsinghua University, Beijing 100084, China. [4]Beijing Frontier Research Center for Biological Structure, Beijing Advanced Innovation Center for Structural Biology, Tsinghua University, Beijing 100084, China. [5]State Key Laboratory of Membrane Biology, Tsinghua University, Beijing 100084, China. [6]These authors contributed equally: Dandan Qian, Ye Cong. ✉e-mail: chenq@nankai.edu.cn; yancy2019@tsinghua.edu.cn; gongds@nankai.edu.cn

encephalomyocarditis virus (EMCV) and herpes simplex virus-1 (HSV-1) showed impaired production of antiviral cytokines and reduced survival after EMCV or HSV-1 exposure[12]. In addition, SIDT2 can directly transport RNA and DNA into lysosomes for degradation in an unexpected ATP-dependent manner[13, 14], which is inconsistent with the characteristic of a channel. Recently, SIDT2 was identified as a sodium-conducting protein in the lysosomal membrane, suggesting that it may act as a sodium/nucleic acid antiporter[15].

However, a contradictory finding has been reported by another group. Luis Vaca and coworkers found that human SIDT1 and SIDT2 share a higher sequence similarity with *C. elegans* tag-130/cholesterol uptake associated protein 1 (CHUP1) that also belongs to SID1 family than with *C. elegans* SID1, suggesting that the mammalian forms function as cholesterol transporters but not as dsRNA transporters due to the existence of two cholesterol recognition/interaction amino acid consensus (CRAC) domains characterized by the presence of the V/L-X(1-5)-Y-X(1-5)-R/K motif[16]. These findings make the functional mechanisms of SID1 family proteins even more perplexing.

Besides nucleic acid transport, SIDT2 also plays important roles in lipid metabolism. SIDT2-deficient mice exhibited an increase in serum triglycerides and free fatty acids[17], a remarkable accumulation of lipid droplets in the liver[18], and changes in lysosomal membrane permeabilization and lipid metabolism[19]. Genome-wide association studies revealed that SIDT2 was associated with high-density lipoprotein cholesterol levels and premature coronary artery disease[20]. In addition, SIDT2 has diverse roles in the regulation of insulin secretion[21–24], lung and gastrointestinal tumor development[25], inflammatory signaling pathways[26], mitochondrial quality control[27], lysosome function[28], autophagy[18,29], and Alzheimer's and Parkinson's disease[30,31]. As the SID1 family shows no striking homology to any known channels or to any transporters, it is still unclear whether it functions as a channel or a transporter. The current lack of any structural information regarding the SID1 family proteins has severely hindered our understanding of their mechanism of action.

In this work, we report the cryo-EM structures of full-length human SIDT2 and characterize its lipid hydrolytic properties.

## Results

### Structural determination of human SIDT2 under three conditions

Previous study has suggested that the mouse SIDT2 is required to transport internalized extracellular dsRNA from endocytic compartments into the cytoplasm[12]. To investigate whether the human SIDT2 can also facilitate this process, we incubated the wild-type (WT) Hela cells and the cells that overexpress human SIDT2 (OE-SIDT2) with extracellular poly(I:C)-rhodamine, respectively, and compared the subcellular localization of dsRNA via confocal microscopy (Supplementary Fig. 1). dsRNA localization in WT cells displayed an obvious punctate accumulation, consistent with an endo-lysosomal accumulation reported by the previous study[12], whereas in OE-SIDT2 cells was predominantly diffuse, consistent with a cytoplasmic distribution reported by the previous study[12]. This result indicates that the human SIDT2 promotes the transport of dsRNA across the endo-lysosomal membrane.

As SIDT2 localizes to the plasma membrane and late endosomal and lysosomal membrane with different pH environments (Fig. 1a)[32,33], Condition 1 (the apo state in pH 7.4, hereafter apoSIDT2-pH 7.4) and Condition 2 (the apo state in pH 5.5, hereafter apoSIDT2-pH 5.5) were used to investigate whether the structure of SIDT2 is affected by pH conditions. Human SIDT1 and SIDT2 share 57% sequence identity and both proteins can transport single-stranded RNA[11,32], double-stranded (ds) RNA[12,34], and dsDNA[14], indicating that SIDT1/SIDT2-mediated transport seems to function independent of the nature of nucleic acids[35]. As SIDT1-mediated plant-derived miRNA2911 absorption in the stomach is low-pH dependent and SIDT2 mainly localizes to the

lysosomes in a low-pH environment, we selected miRNA2911 to serve as the substrate of SIDT2 in our study, which we designed to investigate the gating mechanisms of SIDT2 regulated by RNA (hereafter SIDT2-pH 5.5 plus miRNA).

The detailed protocols of the protein purification, sample preparation, cryo-EM data acquisition, and structural determination are presented in Methods, Supplementary Figs. 2-6, and Supplementary Table 1. The apoSIDT2-pH 7.4, apoSIDT2-pH 5.5, and SIDT2-pH 5.5 plus miRNA structures were determined at overall resolutions of 3.16 Å, 3.21 Å, and 2.87 Å, respectively (Supplementary Figs. 3 and 4 and Supplementary Table 1). No reliable RNA density was observed in the map of SIDT2-pH 5.5 plus miRNA, and no conformational change was found in any of these structures; therefore, we used the SIDT2-pH 5.5 plus miRNA structure with the highest resolution of 2.87 Å for structural analysis. The 2.87 Å EM map displays excellent main chain connectivity and side chain densities for almost all residues of the extracellular/luminal domain (ECD) of SIDT2 (Supplementary Fig. 5a, b). Notably, the conformation of the ECD is almost the same as the ECD predicted by AlphaFold (hereafter AF)[36], which can be docked into the map with minor adjustment. The densities for transmembrane helix (TM) 2 and TM6-9 were obtained at lower resolution (Supplementary Fig. 5c). The cytoplasmic domain (CTD) was invisible in the maps of apoSIDT2-pH 7.4 and apoSIDT2-pH 5.5, reflecting intrinsic flexibility (Supplementary Fig. 3d). An additional unmodeled density was observed on the cytoplasmic side in the map of SIDT2-pH 5.5 plus miRNA (Supplementary Fig. 3d).

### Overall structure of SIDT2

The human *SIDT2* gene encodes an 832-amino acid protein with abundant glycosylation and 8–11 predicted transmembrane segments[35]. Based on our structure, SIDT2 contains a signal peptide, two β-strand rich domains (BRDs), a large unresolved CTD located between TM1 and TM2, and eleven transmembrane segments (Fig. 1a). There were 10 glycosylation sites on SIDT2, according to the database, wherein the glycosylation of N27, N54, N60, N123, N141, and N165 was clearly resolved in the EM map (Fig. 1b, c and 2). The structure revealed a dimeric assembly through an extensive dimer interface that involved both the ECD and transmembrane domain (TMD). The overall structure of SIDT2 had dimensions of approximately 77 Å by 50 Å by 120 Å (Fig. 1b).

### The structures of BRDs

BRD1 residues 23-152 fold into 11 β-strands with 5 glycosylation sites (Fig. 2a, b). The β1, β3, β10, β5, and β6 are packed together and β2, β11, β4, and β9 are well ordered on the opposite side. β7 and β8 are nearly perpendicular to β6 and β9, respectively (Fig. 2a, b). C117 of BRD1 forms a disulfide bond with C207 of BRD2, stabilizing the interface between BRD1 and BRD2 (Fig. 2a, b and Supplementary Fig. 6). BRD2 residues 157-287 fold into 8 β-strands with 1 glycosylation site (Fig. 2a, b). The β1, β8, β3, and β6 are packed together and β2, β7, β4, and β5 are well ordered on the opposite side (Fig. 2c). A disulfide bond is formed between C197 and C256 (Fig. 2c and Supplementary Fig. 6).

From the side view, the electrostatic potential surface shows a large fenestration in the dimer interface (Supplementary Fig. 7a). Two separated tunnels are formed from the fenestration to the bottom of the BRD2 dimer interface (Supplementary Fig. 7a). A large number of positively charged residues are found on both BRD1 and BRD2 (Supplementary Fig. 7b).

### The structure of the TMD

The TMD consists of 11 TM portions. Two pairs of disulfide bonds are found in the extracellular loops, wherein C484 and C490 in the loop between TM2 and TM3 (loop 2-3) pair with C570 in the loop between TM4 and TM5 (loop 4-5) and C787 in the loop between TM10 and TM11 (loop 10-11) (Fig. 2d and Supplementary Fig. 6). Except for TM2, the

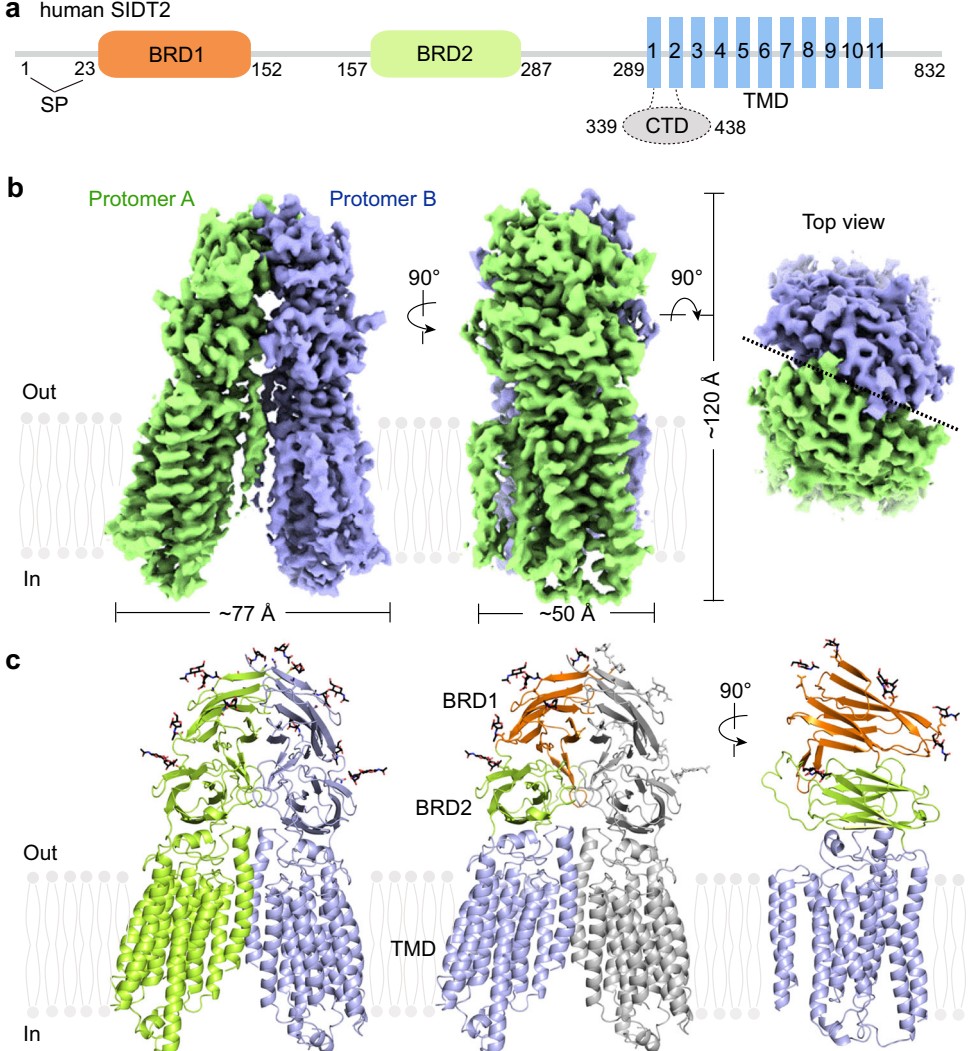

**Fig. 1 | Overall structure of human SIDT2. a** A schematic diagram showing the domain organization of SIDT2. SP signal peptide, BRD β-sheet rich domain, TMD transmembrane domain, CTD cytoplasmic domain. The dashed line indicates the unresolved region in the structure. **b** The overall EM density map of the human SIDT2. SIDT2 is a homodimer, and the protomers A and B are shown in lemon and light blue, respectively. The map was contoured at 0.3 and visualized in ChimeraX (www.cgl.ucsf.edu/chimerax). **c** Overall structure of SIDT2. The N-linked glycans are displayed as black sticks. All structural figures were prepared in PyMOL (www.pymol.org).

TMs are arranged counterclockwise in an orderly manner from the extracellular/luminal view (Fig. 2d).

Remarkably, an ion dense region is found around three histidine residues, H568 in TM4 and H796 and H800 in TM11. In cooperation with S564 in TM4 and D579 in TM5, this motif (H₃-D-S) forms a putative zinc-binding site (Fig.2d and Supplementary Fig. 5d), which was also identified in the structures of adiponectin receptors (ADIPORs)[37] and alkaline ceramidase 3 (ACER3)[38]. The putative $Zn^{2+}$ is located approximately 6 Å from the extracellular/luminal surface of the membrane (Fig. 2d). The two pairs of disulfide bonds in the extracellular loops are situated immediately above the putative $Zn^{2+}$-binding site, maintaining a large space around it (Fig. 2d).

## SIDT2 dimer interface

The dimer interface of SIDT2 can be divided into three regions. The first region is formed by the β1, β3, β10, β5, and β6 of two BRD1 molecules through extensive hydrophobic interactions and one hydrogen bond (Fig. 3a, b). The second region is formed between the loop of β4 and β5 (loop 4-5) in BRD2 of one protomer with loop 4-5 in BRD2 of the opposing protomer and with the loop of β7 and β8 (loop 7-8) in BRD1 of the opposing protomer (Fig. 3a, c). Y210 and F218 in loop 4-5 form cation-π interactions with R100 in loop 7-8 of the opposing protomer. A hydrogen bond is formed between the N215. In addition, D204 in loop 4-5 forms ionic interactions with R100 and K106 of the opposing protomer (Fig. 3a, c). The third region is formed between TM2 of one protomer with TM6 and the extracellular/luminal end of TM5 of the opposing protomer (Fig. 3a, d), and this formation is possibly mainly because of extensive hydrophobic interactions.

## No discernible nucleic acid conduction pathway is found

It has been reported that *C. elegans* SID1 is a dsRNA-gated channel that functions independent of ATP[39], indicating that the SID1 family requires an open pore of more than 20 Å to transport dsDNA or dsRNA. However, no discernible nucleic acid conduction pathway is present in the TMD or TMD dimer interface of SIDT2 (Supplementary Fig. 8a), suggesting that it may act as a transporter. In addition, no conformational change was found between the apoSIDT2-pH5.5 and apoSIDT2-pH7.4 structures, indicating that SIDT2 is insensitive to pH conditions. Notably, the only difference in these three structures is that an additional dense area is observed in the map of SIDT2-pH 5.5 plus miRNA in the region that, on the basis of the AF-SIDT2 structure, was expected to be the CTD (Supplementary Fig. 8).

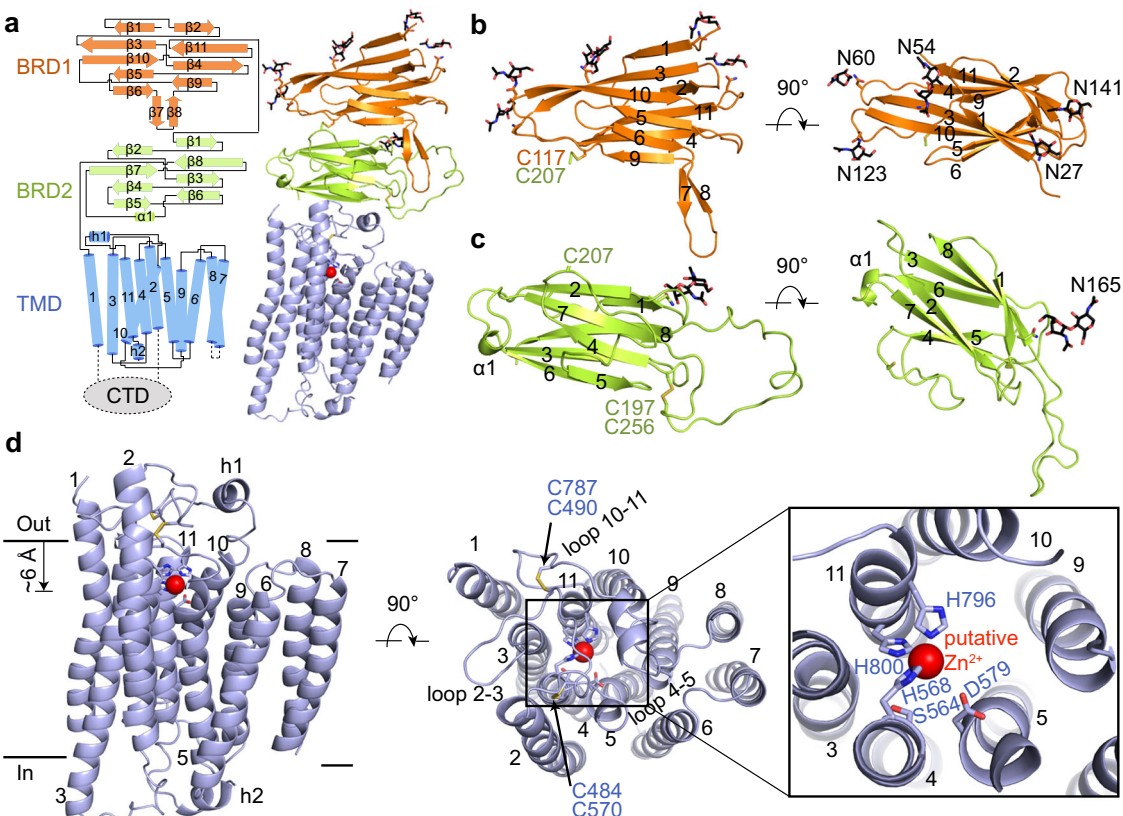

**Fig. 2 | Structural features of each domain. a** Topological diagram and overall structure showing one SIDT2 protomer. The CTD is invisible in the structure, reflecting its flexibility. **b** BRD1 contains eleven β-sheet folds as two lamellar structures facing each other. One disulfide bond is formed between C117 of BRD1 and C207 of BRD2, and five glycosylation sites are observed in BRD1. **c** BRD2 contains eight β-sheet folded into two lamellar structures facing each other. One glycosylation site and one disulfide bond are present. **d** The TMD contains eleven transmembrane helices containing a putative $Zn^{2+}$-binding site formed by three histidine residues, one aspartate residue, and one serine residue. The putative $Zn^{2+}$ is located approximately 6 Å from the extracellular/luminal surface of the membrane. Two pairs of disulfide bonds are formed in the extracellular/luminal loops. Except for TM2, the TMs are arranged counterclockwise in an orderly manner from the extracellular/luminal view. loop 2–3 indicates the loop between the TM2 and TM3, loop 10-11 indicates that the loop between the TM10 and TM11.

This unmodeled area of density blocks the cytoplasmic entry point in the dimer interface (Supplementary Fig. 8).

Although human SITD2 shares an extremely similar structural folding with that of both *C. elegans* AF-SID1 and AF-CHUP1 structures, the conformation of the SIDT2 protomer is strikingly similar to that of CHUP1, with a main-chain RMSD (root mean square deviation) of the intact structure of approximately 1.92 Å, which is comparable to that of AF-SIDT2, whereas the RMSD for AF-SID1 is approximately 12.80 Å (Supplementary Fig. 9a). Superimposing the TMDs of SIDT2, AF-CHUP1, and AF-SID1 on the basis of the $Zn^{2+}$-binding site shows that TM6-9 undergo large conformational changes (Supplementary Fig. 9b). Superimposing the ECDs of SIDT2, AF-CHUP1, and AF-SID1 shows that the ECD of SIDT2 undergoes obvious conformational changes compared to that of AF-SID1, but displays a nearly identical conformation to that of AF-CHUP1 (Supplementary Fig. 9c). These results indicate that the functional mechanisms of SIDT2 may be more similar to those of CHUP1 than to those of SID1. However, even under a cholesterol-mimicking solubilization condition (1% CHS, cholesteryl hemisuccinate), no densities for the CHS or native cholesterol molecules were found around the two CRACs (cholesterol recognition/ interaction amino acid consensus motif) (Supplementary Fig. 9d).

## SIDT2 shows a $Zn^{2+}$-dependent catalytic core similar to that of ACER3 and ADIPOR2

The overall structure of SIDT2 shows no clear characteristic of a channel protein. We performed a structure homology search using DALI, a protein structure comparison server (http://ekhidna2. biocenter.helsinki.fi/dali/)[40] and identified human ACER3 (PDB code: 6G7O)[38], which shared a similar $Zn^{2+}$-dependent catalytic core in the TMD region. Interestingly, the sequence $H_3$-S-D motif is common to members of a superfamily of putative hydrolases identified based on statistically significant sequence similarities, called CREST (ACER, progesterone adipoQ receptor (PAQR) receptor, Per1, SID1, and TMEM8) hydrolases[41].

Superimposing the TMD of SITD2 with that of ACER3 and ADI-POR2 relative to the $Zn^{2+}$-binding sites, TM3-6 and TM9-11 of SIDT2 was clearly aligned with TM1-7 of ACER3 and ADIPOR2 (Fig. 4a). TM5, TM6, and TM9 of SIDT2 showed a marked conformational change compared to TM3-5 of ACER3 and ADIPOR2. In contrast, TM3, TM4, TM10, and TM11 displayed a relatively minor conformational change compared to TM1, TM2, TM6, and TM7 of ACER3 and ADIPOR2, wherein the conserved $H_3$-S-D motif lies (Fig. 4a). In addition, the $Zn^{2+}$-binding sites of ACER3 and ADIPOR2 face the cytoplasm whereas that of SIDT2 faces the extracellular/luminal side (Fig. 4b). A large cavity around the $H_3$-S-D motif was found in both the ACER3 and ADIPOR2, providing space for binding of the ceramide substrates[38, 42]. Similarly, a large cavity connected to the extracellular/luminal hydrophilic environment was observed within the TMs of SIDT2 (Fig. 4b), suggesting that water molecules can have constant access to the active site of SIDT2. Human PGAP3 (post-glycosylphosphatidylinositol (GPI) attachment to proteins phospholipase 3), the ortholog of *S. cerevisiae* Per1p, is required for the hydrolysis of lipid moieties in GPI-anchored proteins[43]. Human myomaker, a member of TMEM8 family, is required for myoblast fusion and muscle formation but it lacks the potentially critical

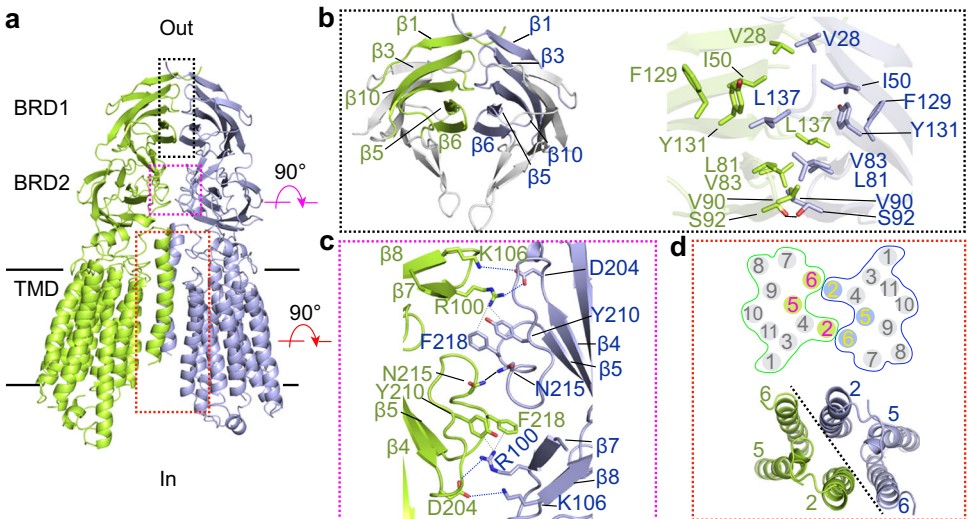

**Fig. 3 | Dimer interface of SIDT2. a** The dimer interface of SIDT2 can be divided into three regions. **b** The first region is formed by the β1, β3, β10, β5, and β6 of the two BRD1 molecules through extensive hydrophobic interactions and one hydrogen bond. The contact hydrophobic residues are indicated. The black dashed line indicates a hydrogen bond. **c** The second region is formed between loop 4-5 of one

protomer with that of the opposing protomer and loop 7-8 of BRD1 of the opposing protomer. The blue dashed lines indicate ionic interactions, and the gray dashed lines indicate cation-π interactions. **d** The third region is formed between TM2 of one protomer and TM6 and the extracellular/luminal end of TM5 of the opposing protomer.

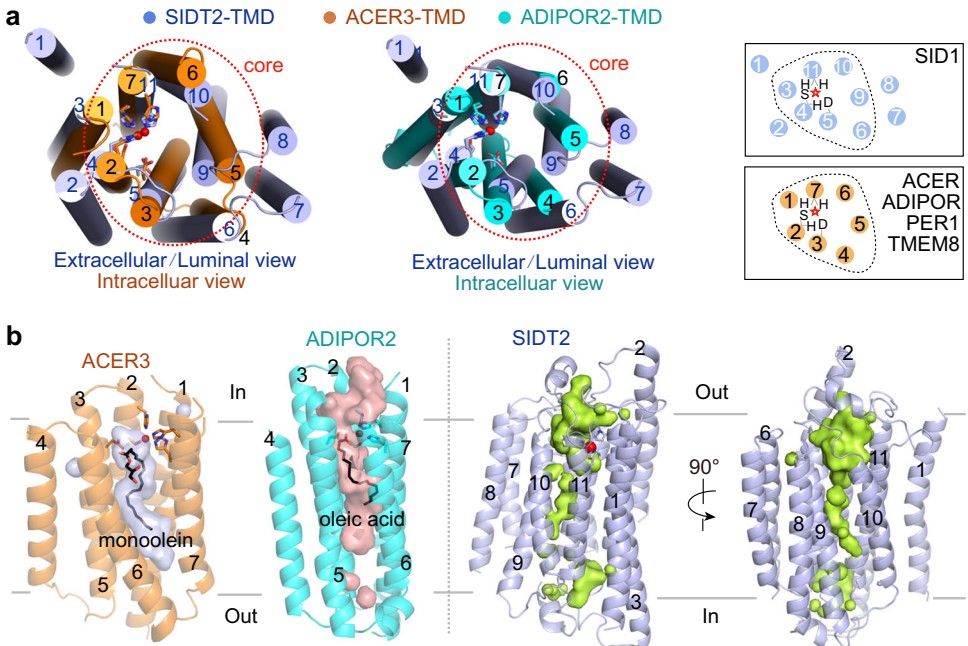

**Fig. 4 | SIDT2 has a Zn²⁺-dependent catalytic core similar to that of ACER3 and ADIPOR2. a** TM3-6 and TM9-10 of SIDT2 are aligned with TM1-7s of ACER3 and ADIPOR2. Superimposing the TMD of SITD2 with that of ACER3 and ADIPOR2 relative to the Zn²⁺-binding sites. **b** A large but distinct cavity around the Zn²⁺-

binding site was observed in all the structures of ACER3, ADIPOR2, and SIDT2. The binding of the lipids in the cavity of the ACER3 and ADIPOR2 structures are indicated as black sticks.

aspartate and serine residues thought to be important for the catalytic activity of hydrolases[41,44]. A large cavity was also found within the TMs of AF-PGAP3, but not in myomaker, which lost hydrolase activity (Supplementary Fig. 10). These results indicate that SIDT2 may exhibit lipid hydrolytic activity.

## SIDT2 exhibits ceramidase activity

To investigate whether SIDT2 exhibits ceramidase activity, ceramide (d18:1/18:0) serves as the substrate of SIDT2 and the products were detected by liquid-chromatography-mass spectrometry. The results

clearly showed that SIDT2 can hydrolyze it into sphingosine (d18:1) and a free fatty acid (C18:0), whereas the H976A-H800A mutant was devoid of this activity (Fig. 5a and Supplementary Figs. 11 and 13). Michaelis-Menten analysis revealed that the SIDT2 has a Michaelis constant ($K_m$) of ~9.0 μM and a catalytic constant ($k_{cat}$) of ~0.12 × 10⁻³ s⁻¹ in our detergent micelles condition (Fig. 5b, Supplementary Fig. 11, and Supplementary Data 1), which is on the same order of magnitude as the $k_{cat}$ for the adiponectin receptor (~0.49 × 10⁻³ s⁻¹), γ-secretase (~1.2 × 10⁻³ s⁻¹), and alkaline ceramidase (~0.66 × 10⁻³ s⁻¹, the $k_{cat}$ is purely an estimate based on the amount of protein in microsomes and

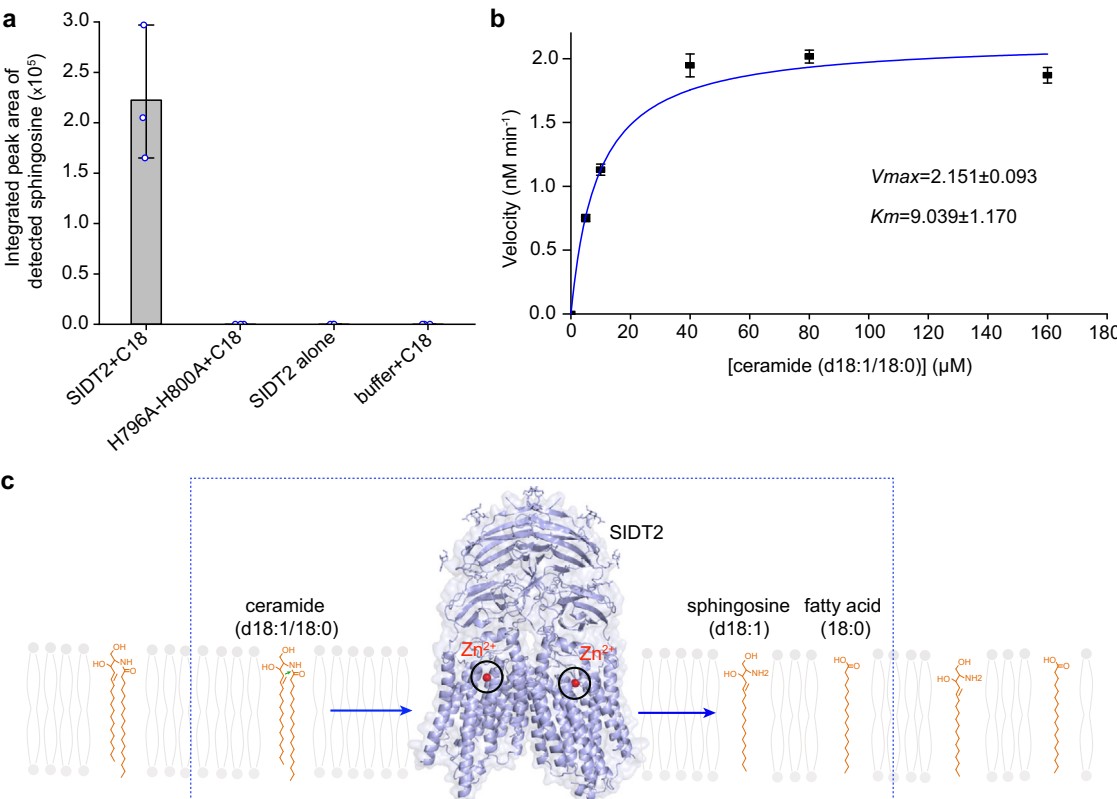

**Fig. 5 | SIDT2 exhibits ceramidase activity. a** The ceramidase activity was detected in the wild-type SIDT2 sample but not in the H796A·H800A mutant sample. The SIDT2 alone condition and the reaction buffer plus ceramide (d18:1/18:0) (C18) condition serve as controls. Detected sphingosine values are represented as the mean ± s.d. (standard deviations) of three independent measurements ($n = 3$). Source data are provided as a Source Data file. **b** Michaelis-Menten analysis of the SIDT2 ceramidase activity. Detected sphingosine values are represented as the mean ± s.d. of three independent measurements ($n = 3$). Source data are provided as a Source Data file. **c** Schematic illustration of the hydrolysis of ceramide by SIDT2.

has not been directly determined)[42], another three kinds of intra-membrane enzymes with very slow rates. In addition, SIDT2 can't hydrolyze the sphingomyelin (d18:1/18:0) (Supplementary Fig. 12), reflecting substrate specificity. Take together, our data suggest that the SID1 family possesses intramembrane hydrolytic activity.

## Discussion

Craig P. Hunter and coworkers firstly found that the *C. elegans* SID1 can passively transport dsRNA without requiring ATP, they proposed that the SID1 acts as a channel[5]. Although this finding rules out three ATP-dependent processes such as primary active transport, endocytosis, and phagocytosis, it cannot rule out the possibilities that it may act as a uniporter, a carrier-mediated process also without requiring energy, and as an antiporter or a symporter, the secondary active transport processes using the energy of an electro-chemical gradient. In contrast, the mammalian SIDT2 was found to transport RNA in an ATP-dependent manner[13] or may function as a sodium/nucleic acid antiporter[15]. As no nucleic acid conduction pathway was found in the TMD in our study, we reasonably speculate that the human SIDT2 may function as a transporter. Due to the low sequence identity between SID1 and SIDT2 and the lack of systemic RNAi in human, suggesting that SID1 and SIDT2 have their respective unique features in structures and functions. Whether the SID1 can form a channel remains to be investigated. Although SIDT2 shares high structural similarity with CHUP1, whether SIDT2 can transport cholesterol remains to be addressed. Structures of SID1, CHUP1, and SIDT2 in more conformations are necessary to fully understand their respective functional mechanisms.

An additional area of density with extremely low resolution was observed in the map of SIDT2-pH 5.5 plus miRNA, we cannot reliably

distinguish whether it belongs to the protein or the RNA. Based on the structure of AF-SIDT2, a portion of the CTD was predicted to be located in the dimer interface (Supplementary Fig. 8), suggesting that the additional area of density is more likely a portion of the CTD. It has been reported that an arginine-rich motif in the CTD is capable of binding nucleic acids[45]. As the CTD is unresolved in the maps of SIDT2 alone samples, reflecting its intrinsic flexibility, the binding of RNA may stabilize this region.

A previous study showed that the recombinant ECD of human SIDT1 forms a tetramer[46]. In our study, the full-length SIDT2 protein was expressed and the structure revealed that the two TMDs also play important roles in forming a dimer, which may be the main reason for this discrepancy.

The closed ADIPOR2 structure favored the binding of an oleic acid, an intermediate ACER3 structure bound a monoolein, and an open ADIPOR1 structure without a ligand suggested that the shifts of TM4 and TM5 may be related to distinct steps of a common catalytic process (Supplementary Fig. 14a)[38]. Superimposing the TMD of SITD2 with that of AF-SIDT2 revealed that TM6-9 and the loop between TM10 and helix 1 (h1) (hereafter loop 10-h1) underwent dramatic conformational changes, whereas the other TMs remains nearly unchanged (Supplementary Fig. 14b). For instance, compared to that of SIDT2, the L770 in the loop 10-h1 of AF-SIDT2 extended outward relative to TM10, by 7 Å. The S716, representing TM9, moved outward relative to TM10, by 3 Å. M709, representing for the TM8, moved outward relative to the TM10 by 3 Å, resulting in a greater opening between TM8/TM9 with TM10 (Supplementary Fig. 14b). Accordingly, a larger cavity was found within the TMD of AF-SIDT2 than in SIDT2 (Supplementary Fig. 14c). The two distinct conformations might represent distinct functional states, reflecting the importance of TM6-9. Further studies will be required to

characterize the precise and diverse functional mechanisms of SIDT2 in lipid metabolism and nucleic acid transport. Taken together, the data obtained in our study mark an important step toward the elucidation of the functional mechanisms of the SID1 family proteins.

## Methods

### dsRNA subcellular localization

The procedures for characterizing the subcellular localization of poly(I:C)-rhodamine (InvivoGen), a synthetic analog of dsRNA, were similar to previously described procedures[47]. The wild-type Hela cells and the cells that overexpression of SIDT2-FLAG proteins were incubated with 1 μg/ml poly(I:C)- rhodamine for 24 h, washed by PBS (phosphate-buffered saline) for three times and treated with 100 μg/ml RNase A enzyme, then fixed by 4% paraformaldehyde (PFA) at 37 °C for 15 min and incubated with FLAG-M2 primary antibody (1:1000, sigma), anti-mouse Alexa-488 secondary antibody (1:1000, ThermoFisher), and DAPI, and finally imaged on a confocal microscope. The dsRNA intensity and area were calculated by the FIJI software package[48].

### Transient protein expression and purification

The full-length human *SIDT2* cDNA was subcloned into the pCAG vector with a C-terminal FLAG-tag and C-terminal His$_8$-tag. HEK293F cells (Invitrogen) were cultured in SMM 293T-II medium (Sino Biological Inc.) at 37 °C under 5% $CO_2$ in a Multitron-Pro shaker (Infors, 130 rpm). When the cell density reached $2.0 \times 10^6$ cells per ml, the pCAG-SIDT2 plasmids were transiently transfected into the cells. For 1-litre HEK293F cell culture, ~2 mg of plasmids were pre-mixed with 4.0 mg 25-kDa linear polyethylenimines (PEIs) (Polysciences) in 50 ml fresh medium for 20–30 min before transfection. The 50 ml mixture was then added to the cell culture. The transfected cells were cultured for 48 h before harvesting.

For purification, 12 l cells were harvested by centrifugation at 800 g for 10 min and resuspended in the lysis buffer containing 25 mM HEPES pH 7.4 and 150 mM NaCl (lysis buffer A), 1.95 μg/ml aprotinin, 1.5 μg/ml pepstatin, and 3 μg/ml leupeptin. The lysate was incubated in the buffer containing 1% (w/v) decyl maltose neopentyl glycol (DMNG) (Anatrace) and 0.1% (w/v) cholesterol hemisuccinate (CHS) (Anatrace) at 4 °C for 2 h for membrane protein extraction. After ultracentrifugation at 18,700 × $g$ for 1 h, the supernatant was collected and applied to the anti-FLAG M2 affinity gel (Sigma) at 4 °C for one time. The resin was washed six times with 5 ml wash buffer A (lysis buffer A plus 0.01% GDN). The protein was eluted with elution buffer A (wash buffer A plus 300 μg/ml FLAG peptide (Sigma)). The eluent was incubated with nickel affinity resin (Ni-NTA, Qiagen) at 4 °C for 50 min, the resin was washed with wash buffer B (lysis buffer A plus 0.01% GDN and 30 mM imidazole), and the protein was eluted with elution buffer B (lysis buffer A plus 0.01% GDN and 300 mM imidazole). The eluent was concentrated and subjected to size-exclusion chromatography (SEC, Superose 6 Increase, 10/300, GE Healthcare) in a buffer containing 25 mM HEPES pH 7.4, 150 mM NaCl, and 0.006% GDN. The peak fractions were pooled and concentrated to ~13 mg/ml for the cryo-EM analysis.

For preparation of the SIDT2 in pH5.5, the protocol is the same as above mentioned only except changing the gel filtration buffer to MES pH 5.5 in the last step. For preparation of the SIDT2-pH 5.5 plus miRNA sample, the RNA molecules (5-ggccggggggacgggcuggga-3) and SIDT2-pH 5.5 proteins were mixed with a mole ratio of 5:1. The expression and purification of the H797A-H800A mutant were the same as for the wild-type SIDT2.

### Cryo-EM data acquisition

Holey carbon grids (Quantifoil Au 300 mesh, R1.2/1.3) were glow-discharged in the Plasma Cleaner PDC-32G-2 (Harrick Plasma Company) with a vacuum for 2 min and mid force for 30 s. Aliquots (4 μl) of SIDT2 proteins were placed on the glow-discharged grids, which were then blotted for 3 s and flash frozen in liquid

ethane cooled by liquid nitrogen using Vitrobot Mark IV (Thermo Fisher Scientific) at 8 °C and 100% humidity. The grids were loaded onto a 300 kV Titan Krios (Thermo Fisher Scientific Inc.) equipped with K3 Summit detector (Gatan) and GIF Quantum energy filter. Images were automatically collected using AutoEMation[49] in super-resolution mode at a nominal magnification of 81,000 × (64,000 × for apoSIDT2-pH 7.4 dataset), with a slit width of 20 eV on the energy filter. A defocus series ranging from −1.3 μm to −1.8 μm was used. Each stack was exposed for 2.56 s with an exposure time of 0.08 s per frame, resulting in a total of 32 frames per stack and the total dose was approximately 50 e-/Å$^2$ for each stack. The stacks were motion corrected with MotionCor2[50] and binned 2 fold, resulting in a pixel size of 1.0825 Å/pixel (1.0979 Å for apoSIDT2-pH 7.4 dataset). Meanwhile, dose weighting was performed[51]. The defocus values were estimated with Gctf[52].

### Image processing

Dose-weighted micrographs were used for contrast transfer function (CTF) estimation using Patch-CTF in cryoSPARC[53]. Micrographs with CTF fitting resolution worse than 3.6 Å were excluded during manual curation. Initial particles were picked from good micrographs using blob picker in cryoSPARC[53] and 2D averages were generated. Final particle picking was done by template picker using templates from the 2D results. Particles were extracted with a box size of 256 pixels and cropped into 128 pixels to accelerate early-step calculation and the yielded particles were re-extracted for final refinement.

For apoSIDT2-pH 7.4 dataset, a previous dataset with 4141 micrographs was collected but failed to reach the atomic resolution. Selected 2D averages were used for template-based picking and ab-initio reconstruction (Supplementary Fig. 3). After optimization of the sample, a total of ~3081k particles were extracted from a new dataset (5170 micrographs) with a pixel size of 2.1958 Å. After several rounds of heterogeneous refinement for guided multi-reference 3D classification, a ~4.5 Å map with clear transmembrane helices and extracellular domain was generated. The remaining ~660k particles were input as seeds to conduct seed-facilitated 3D classification[54]. After that, ~655k particles were re-extracted with a pixel size of 1.0979 Å. Ab-initio reconstruction for non-reference 3D classification and heterogeneous refinement for guided multi-reference 3D classification were carried out on these particles alternatively, yielding ~184k good particles. These particles were subjected to non-uniform refinement and local refinement with C2 symmetry, generating a final map with an overall resolution of 3.16 Å.

A same protocol was performed for processing the apoSIDT2-pH 5.5 and SIDT2-pH 5.5 plus miRNA datasets. A total of ~1,157k (~2152k for SIDT2-pH 5.5 plus miRNA dataset) particles were extracted from 2192 (3676 for SIDT2-pH 5.5 plus miRNA dataset) micrographs with a pixel size of 2.165 Å. The 3.16 Å map of the apoSIDT2-pH 7.4 was low-pass filtered to 6 Å to serve as the initial model. After several rounds of heterogeneous refinement, non-uniform refinement, and local refinement, the final reconstruction reached 3.21 Å (2.87 Å for SIDT2-pH 5.5 plus miRNA dataset).

### Model building and structure refinement

An initial structure model for SIDT2 was generated by AlphaFold[36]. The predicted ECD and TMD structures were docked into the density map and manually adjusted and re-built by COOT[55], respectively. Sequence assignment was guided mainly by bulky residues such as Phe, Tyr, Trp, and Arg. Unique patterns of sequences were exploited for validation of residue assignment. For the ECD, the glycosylation sites and disulfide bonds also facilitate sequence assignment. Structure refinements were carried out by Phenix in real space with secondary structure and geometry restraints[56]. The statistics of the 3D reconstruction and model refinement are summarized in Supplementary Table 1.

## Ceramidase activity assay

The enzymatic activities of the wild-type SIDT2 and H796A-H800A mutant were probed using liquid-chromatography-mass spectrometry (LC-MS) analyses. Purified SIDT2 (0.6 μM) proteins were incubated with ceramide (d18:1/18:0) (40 μM) or sphingomyelin (d18:1/18:0) (40 μM) (Avanti Polar Lipids) for 3 h at room temperature in 25 mM Tris pH 8.0, 150 mM NaCl, 0.02% (w/v) DDM, and 0.002% (w/v) CHS. The reactions were stopped by addition of methanol (30% final). With regard to the H796A-H800A mutant, the same reaction system was adopted and the ceramide (d18:1/18:0) acts as the substrate. With regard to the Michaelis-Menten analyses, purified SIDT2 proteins (0.3 μM) were incubated for thirty minutes at room temperature with increasing amounts of ceramide (d18:1/18:0) substrate (5, 10, 40, 80, 160 μM). The reactions were stopped and analyzed as described above. Each experiment was performed in triplicate. Data were fitted to the Michaelis-Menten equation using Origin software. The D-erythro-sphingosine (d18:1) (Avanti Polar Lipids) serves as the standard sample. The ceramidase activity was quantified by peak area comparison with sphingosine standards. In each condition, hydrolyzed substrate represented less than 1% of the total substrate concentration. The SIDT2 alone condition and the reaction buffer plus ceramide (d18:1/18:0) condition serve as controls. Lipids were extracted from reaction samples using the previously reported method[38]. Briefly, a mixture of dichloromethane/methanol/water (2.5:2.5:2 v/v/v) was added to the reaction and the solution was centrifuged. The organic phase was collected and dried under nitrogen, then dissolved in 35 μl of methanol. The lipid extract was stored at −20 °C before LC-MS analysis.

LC-MS analysis was performed using a Waters ACQUITY UPLC I-Class. The samples were separated on an Acquity UPLC BEH-C8 column (particle size 1.7 μm, 2.1 × 50 mm) (Waters) maintained at 35 °C. The mobile phases consisted of eluent A (0.1% formic acid) and eluent B (0.1% formic acid-acetonitrile). The gradient was as follows: 97% A plus 3% B at 0 min, 97% A plus 3% B at 2 min, 100% B at 7 min, 100% B at 9 min, and 97% A plus 3% B at 9.1 min. The flow rate was 0.4 ml/min. The auto sampler was set at 5 °C and the injection volume was 10 μl. The HPLC system was couple d on-line to a SYNAPT G2-SI MS equipped with electrospray ionization source operated in positive ion mode. The source parameters used were as follows: source temperature was set at 100 °C, cone gas flow rate was 30 L/h, desolvation gas flow rate was 700 L/h.

## Statistical analysis

Statistical analyses were performed by GraphPad (https://www.graphpad.com/quickcalcs/ttest2/) using two-tailed Student's t-tests.

## Reporting summary

Further information on research design is available in the Nature Portfolio Reporting Summary linked to this article.

## Data availability

The atomic coordinates of apoSIDT2-pH 7.4, apoSIDT2-pH 5.5, and SIDT2-pH 5.5 plus miRNA have been deposited in the PDB (http://www.rcsb.org) under the accession code 7Y63, 7Y69 and 7Y68, respectively. The electron microscopy density maps of these three structures have been deposited in the Electron Microscopy Data Bank (EMDB https://www.ebi.ac.uk/pdbe/emdb/) under the accession code EMD-33632, EMDB: EMD-33638, and EMD-33637, respectively. The source data underlying Fig. 5a, b, and Supplementary Figs. 1b, 1c, 2a, and 2b are provided in the Source Data file. Source data are provided with this paper.

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

## Acknowledgements

We thank the Tsinghua University Branch of China National Center for Protein Sciences (Beijing) for providing the cryo-EM facility support; the computational facility support on the cluster of Bio-Computing Platform (Tsinghua University Branch of China National Center for Protein Sciences Beijing). We thank Ning Liu in the Mass Spectrometry Platform of the State Key Laboratory of Medicinal Chemical Biology for technical support for LC-MS analysis. This work was supported by funds from the National Natural Science Foundation of China (32271254 to D.G.) and the Fundamental Research Funds for the Central Universities of NanKai University (63223041 to D.G.).

## Author contributions

D.G. conceived the project. Q.C., C.Y., and D.G. supervised the project and designed all experiments. D.Q., Y.C., and R.W. performed the experiments. All authors contributed to the data analysis. D.G. wrote the manuscript.

## Competing interests

The authors declare no competing interests.
