## [Peer Review File · Nature Communications]

Structural insight into the human SID1 transmembrane family member 2 reveals its lipid hydrolytic activityREVIEWER COMMENTS

Reviewer #1 (Remarks to the Author):

The manuscript by Qian et al presents the structure of human SIDT2, a protein annotated as a homolog of the systemic RNA interference-deficient 1 (SID1) family of proteins that are implicated in RNA/DNA transport across cellular membranes. The cryoEM structures identify the general features of this protein, which include extracellular B-sheet-rich domains that contain a small pore with positively charged residues, and a transmembrane domain with 11 alpha helices that shares homology with the CREST family of enzymes (alkaline ceramidases and adiponectin receptors) that are lipid hydrolases. The putative active of SIDT2 is located on the extracellular face of the membrane and contains three histidine residues and a serine residue that are conserved in this enzyme family. The authors note a lack of structural homology between SIDT2 and SID1 based on the alphafold predictions of SID1 and also note that SIDT2 does share structural homology with CHUP1, a cholesterol uptake protein from *C. elegans*, which was recently identified to share higher sequence homology. The lack of structural homology with SID1 and a lack of a clear channel for transport calls into question the role of SIDT2 in RNA/DNA transport. Consequently, the authors propose SIDT2 is a lipid hydrolase and they provide some evidence that SIDT2 can hydrolyze the lipid ceramide. Overall the work is impactful and of broad interest. However, there are a few major concerns that need to be addressed to fully support the new hypothesis that SIDT2 is a lipid hydrolase.

Major points to address

1. How did the authors identify the ion as a Zinc ion? If there is no experimental evidence, it should be labeled a putative zinc ion, or experimental evidence should be provided as to the identity of the ion.
2. Can the authors rule out that the observed hydrolysis of ceramide by SIDT2 is from a contaminating protein? Proper controls would rule this out (e.g. a point mutant of a conserved residue(s) that lacks activity)
3. The authors should quantitate the amount of ceramide hydrolyzed by SIDT2 as a function of protein amount (e.g. either per mol or per ng) and per time (e.g. per second or minute). Given there is no evidence that SIDT2 can affect lipid levels in cells, then the rate should be determined to provide some context whether ceramide hydrolysis by SIDT2 is biologically relevant.
4. Lastly, there are several potential lipid substrates for SIDT2 that contain amide bonds that are not ceramide, which is mainly found in intracellular membrane compartments. Given the active site of SIDT2

faces the extracellular space, the ability of SIDT2 to hydrolyze other lipids abundant on the outer plasma membrane leaflet are more likely candidates for SIDT2 substrates. For example, sphingomyelin, cerebroside, or gangliosides, which have different polar headgroups with a ceramide lipid core. To clarify if SIDT2 is a broad or narrow specificity lipid hydrolase, it would be ideal if the authors compared the rate of sphingomyelin, glucosylceramide, and galactosyl ceramide hydrolysis to the rate of ceramide hydrolysis.

Minor points:

In the background introduction, there are many sentences that say SIDT2 can transport RNA and DNA. Can the authors clarify the exact findings more clearly? For example, has the gene or protein product of SIDT2 been shown to be necessary in cell culture for RNA and DNA transport? Or has SIDT2 been shown to directly transfer RNA and/or DNA across membrane bilayers? Given the authors findings, this may help clarify the reading of this manuscript for those less familiar with SIDT2 function.

I would suggest removing the word “unfortunately” from line 124

Reviewer #2 (Remarks to the Author):

The authors present the cryoEM structure of human SIDT2, a putative nucleic acid transporter channel. The structure reveals a tight dimer that is mediated by interactions between the extracellular beta structure regions and the helical transmembrane region. Surprisingly, there is no channel apparent in the protein appropriate for transport of small double stranded nucleic acids. Instead, the authors find a cavity emanating from the extracellular side towards a buried Zn ion that appears to be able to hydrolyze ceramides, suggesting that SIDT2 is likely a lipid hydrolase.

Overall, the structure is novel, the work is soundly done, and the paper is fairly well written, although it is largely descriptive in nature. However, more experiments are needed to support the structure. Is it possible that the observed dimer is not the physiological oligomeric state and the exposed basic residues in the extracellular beta-domains form the walls of channel in a larger assembly? With the structure now in hand, have the authors considered mutating these basic residues to assess whether that impairs nucleic acid transport function in cellular assays? Without these types of accompanying functional assessments, there is not much advancement of our functional understanding of this protein other than that it may function in lipid hydrolysis. More importantly, how do the authors reconcile the findings here with the many papers that show SIDT2 has nucleic acid transporting capability? It would be worthwhile to include some statements in the discussion about the reasons for the previous

inconsistencies on this protein's role, if possible. In this regard, the paper would benefit from additional cell-based experiments to either test for nucleic acid transport function and/or verify ceramidase function.

Other major points:

1) Mutagenesis of key interface and putative catalytic residues and assessment in a cell-based assays, or at a minimum, measuring the ability of mutants to hydrolyze lipids in vitro would strengthen the paper.

2) Is the cavity next to the Zn consistent with binding a hydrophobic molecule such as ceramide?

Can the authors speculate on the enzymatic mechanism of ceramide hydrolysis based on the structure of the active site?

3) Fig. 5a: where is the ceramide 1+ peak?

4) Extended data Fig. 1a: The purified SIDT2 does not appear to be very pure as there are several significant bands above and below SIDT2. Could these contaminants play a role in the hydrolysis experiments? Mutagenesis of key catalytic residues would clarify this.

5) How do the authors explain the additional density appearing in the SIDT2 structure with added RNA? Presumably the RNA addition is the only difference with the other samples. Could RNA binding be stabilizing the C-terminal region?

Minor points:

1) All occurrences of 'systematic RNAi', presumably should be 'systemic RNAi'

2) L151 – 152: 11 b-strands instead of 11 b-sheets

3) L156: BRD1 and BRD2

4) L152 and L158: What is meant by lamellar?

5) Fig. 3b and 3c: the labels are occluding the structural details; they should be moved away from structure and/or use thin arrows from label to residue.

Reviewer #3 (Remarks to the Author):

The SID1 family proteins are putative nucleic acids channels. In this study the authors revealed the cryo-electron microscopy structures of human SIDT2 (SID1 family). SIDT2 formed a dimer, and nucleic acid conduction pathway was not identified in the transmembrane domain. SIDT2 hydrolyzed C18 ceramide into sphingosine and a free fatty acid.

Although the cryo-electron microscopy structures of SIDT2 is novel, there are multiple major concerns.

1. The significance of this study is weak. The authors could not show the structural basis for nucleic acids transport by SIDT2. The relationship among the SIDT2 structure, nucleic acids transport activity, and ceramidase activity needs to be investigated.

2. To clarify the relationship between structure and activity, whether or not the SIDT2 protein used in this study possesses an RNA transport activity needs to be shown. For example, does the SIDT2 construct enhance the import of extracellular RNA into the cells when the protein is overexpressed in cells?

3. The possibility that the ceramidase activity is due to contaminated proteins is not excluded. To show that SIDT2 indeed exhibits ceramidase activity, the activity of SIDT2 mutant(s) needs to be investigated. For example, mutation(s) on the zinc-binding site (H3-D-S) may interfere with the ceramidase activity.

4. It is unclear whether ceramide is a physiological and main substrate of SIDT2. Does SIDT2 hydrolyze ceramide in cells? Is the ceramidase activity of SIDT2 stronger or weaker than other known ceramidases?

The ceramidase activity of SIDT2 can be predicted easily, because SIDT2 has a homology to other ceramidases (Pei et al. *Biology Direct* 2011, 6:37). Hydrolase activity for other lipids should also be investigated.

5. Pratt et al. reported that the extracellular domain of SIDT1 forms a stable tetramer (Pratt et al. PLoS ONE 2012 e33607). Please explain the discrepancy between the current study and this paper.

Minor point

Fig. 5c. The illustration is wrong. Ceramide, sphingosine and fatty acids should not penetrate a lipid bilayer, but should localize to the inner or outer layer.

Response to reviewers:

A brief summary of major revisions:

We thank these three reviewers for their time and constructive comments! Before addressing the specific concerns from each reviewer, we would like to summarize the major revisions we have made to the manuscript.

1. In the previous version, the expression “putative nucleic acid channel” causes some confusion about the relationship between the structure and function of SIDT2. Actually, although SID1 family proteins are required to transport nucleic acids, it’s unclear that whether SID-1 family proteins function as a channel or a transporter due to the lack of a direct biochemical transport assay in vitro. In addition, the SID1 family proteins were named both dsRNA transporter and dsRNA channel in the previous literatures. In the original paper that named it as a channel (*Science*. 2003; 301:1545-1547), Hunter and colleagues revealed that the *C. elegans* SID1-mediated a passive transport process without consumption of ATP, ruling out internalization of dsRNA by pumps, endocytosis, or phagocytosis. However, it cannot rule out the possibilities that it may act as a uniporter, a carrier-mediated process also without requiring energy, and as an antiporter or symporter, the secondary active transport processes using the energy of an electro-chemical gradient. In fact, a previous study indicates that the mammalian SIDT2 may be a Na⁺/nucleic acid antiporter (*FEBS Lett.* 2017; 591: 76-87.). Also, it has been reported that SIDT2-mediated uptake of RNA for degradation in lysosome in an unexpected ATP-dependent manner (*Autophagy*. 2016; 12: 565–578.), which is inconsistent with the feature of a channel.

Due to the low sequence identity between SID1 and SIDT2 and the lack of systemic RNAi in human, suggesting that SID1 and SIDT2 have respective unique features in structures and functions. In our study, no discernible nucleic acid conduction pathway has been identified within the TMD or the dimerized TMDs, suggesting that SIDT2 seems to be a transporter. Whether the SID1 can form a channel and the underlying mechanisms for transporting nucleic acids by SID1 family proteins need to be further investigated by extensive studies.

For clarity, we have revised the Title, Abstract, Introduction, and Discussion, please see the highlighted paragraphs in these sections.

2. We have performed cell-based dsRNA intracellular localization assay and confirmed that our SIDT2 protein is also required for transporting dsRNA (Supplementary Fig. 1).

3. The lack of ceramidase activity of the H796A-H800A mutant indicates that the wild-type SIDT2 has lipid hydrolytic activity (Fig. 5a and Supplementary Fig. 11). In addition, Michaelis-Menten analysis revealed that the SIDT2 has a catalytic constant (k_{cat}) of $\sim 0.12 \times 10^{-3} \text{ s}^{-1}$ in our detergent micelles condition (Fig. 5b and Supplementary Fig. 12), which is on the same order of magnitude as the k_{cat} for the adiponectin receptor ($\sim 0.49 \times 10^{-3} \text{ s}^{-1}$), γ -secretase ($\sim 1.2 \times 10^{-3} \text{ s}^{-1}$), and alkaline ceramidase ($\sim 0.66 \times 10^{-3} \text{ s}^{-1}$), another three kinds of intramembrane enzymes with very slow rates. In addition, SIDT2 can't hydrolyze the sphingomyelin (d18:1/18:0) (Supplementary Fig. 13), reflecting its substrate specificity.

Response to Reviewers' Comments:

Reviewer #1:

Major points

1. 1. How did the authors identify the ion as a Zinc ion? If there is no experimental evidence, it should be labeled a putative zinc ion, or experimental evidence should be provided as to the identity of the ion.

Good advice. We have revised it as a putative zinc ion.

2. Can the authors rule out that the observed hydrolysis of ceramide by SIDT2 is from a contaminating protein? Proper controls would rule this out (e.g. a point mutant of a conserved residue(s) that lacks activity).

We thank this reviewer for the insightful comments. To consolidate the ceramidase activity of SIDT2, we have generated a H796A-H800A mutant that each residue is crucial for binding the putative zinc ion. Expectedly, the mutant was devoid of the ceramidase activity. In addition, a “reaction buffer plus ceramide (d18:1/18:0)” sample and a “SIDT2 alone” sample act as controls (Fig. 5a and Supplementary Fig. 11).

3. The authors should quantitate the amount of ceramide hydrolyzed by SIDT2 as a function of protein amount (e.g. either per mol or per ng) and per time (e.g. per second or minute). Given there is no evidence that SIDT2 can affect lipid levels in cells, then the rate should be determined to provide some context whether ceramide hydrolysis by SIDT2 is biologically relevant.

Point well taken. We have quantitated the amount of ceramide hydrolyzed by SIDT2 and analyzed by Michaelis-Menten equation (Fig. 5b and Supplementary Fig. 12 and the last paragraph, Page 13), the catalytic constant of SIDT2 in our detergent micelles condition is comparable to those for adiponectin receptor, alkaline ceramidase, and γ -secretase, another three kinds of intramembrane enzymes with very slow rates.

Actually, it has been reported that SIDT2 plays important roles in lipid metabolism. SIDT2-deficient mice exhibited an increase in serum triglycerides and free fatty acids (*Biochem Biophys Res Commun.* 2016; 476:326-332), a remarkable accumulation of lipid droplets in the liver (*J Lipid Res.* 2018; 59:404-415.), and changes in lysosomal membrane permeabilization and lipid metabolism (*Exp Ther Med.* 2018; 16:246-252). Genome-wide association

studies revealed that SIDT2 was associated with high-density lipoprotein cholesterol levels and premature coronary artery disease (*Arterioscler Thromb Vasc Biol.* 2021; 41:2494-2508.) (The first paragraph, Page 5). Taken together, the ceramide hydrolysis by SIDT2 may be biologically relevant.

4. Lastly, there are several potential lipid substrates for SIDT2 that contain amide bonds that are not ceramide, which is mainly found in intracellular membrane compartments. Given the active site of SIDT2 faces the extracellular space, the ability of SIDT2 to hydrolyze other lipids abundant on the outer plasma membrane leaflet are more likely candidates for SIDT2 substrates. For example, sphingomyelin, cerebrosides, or gangliosides, which have different polar headgroups with a ceramide lipid core. To clarify if SIDT2 is a broad or narrow specificity lipid hydrolase, it would be ideal if the authors compared the rate of sphingomyelin, glucosylceramide, and galactosyl ceramide hydrolysis to the rate of ceramide hydrolysis.

We appreciate this excellent suggestion. As several lipids above mentioned are not currently available in our country, we have only obtained sphingomyelin. We found that SIDT2 cannot hydrolyze it, reflecting its substrate specificity (Supplementary Fig. 13).

Minor points

1. In the background introduction, there are many sentences that say SIDT2 can transport RNA and DNA. Can the authors clarify the exact findings more clearly? For example, has the gene or protein product of SIDT2 been shown to be necessary in cell culture for RNA and DNA transport? Or has SIDT2 been shown to directly transfer RNA and/or DNA across membrane bilayers? Given the authors findings, this may help clarify the reading of this manuscript for those less familiar with SIDT2 function.

We appreciate these excellent suggestions. Please also see the first point in the “A brief summary of major revisions” section in Page 1 of this file. We have revised these issues throughout the revised manuscript.

2. I would suggest removing the word “unfortunately” from line 124.

Point taken! We have removed it.

Reviewer #2:

Major points

1. Is it possible that the observed dimer is not the physiological oligomeric state and the exposed basic residues in the extracellular beta-domains form the walls of channel in a larger assembly? With the structure now in hand, have the authors considered mutating these basic residues to assess whether that impairs nucleic acid transport function in cellular assays? Without these types of accompanying functional assessments, there is not much advancement of our functional understanding of this protein other than that it may function in lipid hydrolysis. More importantly, how do the authors reconcile the findings here with the many papers that show SIDT2 has nucleic acid transporting capability? It would be worthwhile to include some statements in the discussion about the reasons.

We thank this reviewer for the insightful comments. Please also see the first point in the “A brief summary of major revisions” section. So far, it is unclear that whether the SID1 family proteins function as a channel or a transporter. According to the literature, SIDT2 seems to be a transporter. Our structure also indicates that SIDT2 may act as a transporter but not a channel. Our data are not only consistent with the previous studies, but also marks an important step towards elucidating the molecular mechanisms of this important SID1 family proteins with varied functions.

For the first question, we did not find any other oligomeric state in the purification step (Supplementary Fig. 2) and 2D or 3D classification step. However, whether the *C. elegans* SID1 can form a channel remains to be determined.

For the second question, we have performed cell-based dsRNA intracellular localization assay and confirmed that our SIDT2 protein is also required for transporting dsRNA (Supplementary Fig. 1). As the lack of the structure of SIDT2 in complex with RNA, it's a big challenge to reveal the recognition and functional mechanism by mutating the basic residue one by one. Accordingly, we are mainly focusing on revealing the ceramidase properties of SIDT2 in this work, the underlying mechanisms for transporting nucleic acid need further extensive studies.

For clarity, we have revised these issues in the Title, Abstract, Introduction, and Discussion (highlighted paragraphs).

2. for the previous inconsistencies on this protein's role, if possible. In this regard, the paper would benefit from additional cell-based experiments to either test for nucleic acid transport function and/or verify ceramidase function.

We thank this reviewer for the insightful comments. We have performed cell-based dsRNA intracellular localization assay and confirmed that our SIDT2 protein is also required for transporting dsRNA (Supplementary Fig. 1). We have also investigated the ceramidase properties of SIDT2 (Fig. 5 and Supplementary Figs. 11-13), revealing that it has a comparable rate to the those of adiponectin receptor, γ -secretase, and alkaline ceramidase, another three kinds of intramembrane enzymes with very slow rates.

In addition, it has been reported that SIDT2 plays important roles in lipid metabolism. SIDT2-deficient mice exhibited an increase in serum triglycerides and free fatty acids (*Biochem Biophys Res Commun.* 2016; 476:326-332), a remarkable accumulation of lipid droplets in the liver (*J Lipid Res.* 2018; 59:404-415.), and changes in lysosomal membrane permeabilization and lipid metabolism (*Exp Ther Med.* 2018; 16:246-252). Genome-wide association studies revealed that SIDT2 was associated with high-density lipoprotein cholesterol levels and premature coronary artery disease (*Arterioscler Thromb Vasc Biol.* 2021; 41:2494-2508.) (The first paragraph, Page 5). Taken together, the ceramide hydrolysis by SIDT2 may be biologically relevant.

3. Mutagenesis of key interface and putative catalytic residues and assessment in a cell-based assays, or at a minimum, measuring the ability of mutants to hydrolyze lipids in vitro would strengthen the paper.

We thank this reviewer for the insightful comments. We have generated a H796A-H800A mutant that each residue is crucial for binding the putative zinc ion. Expectedly, the mutant was devoid of the ceramidase activity (Fig. 5a and Supplementary Fig. 11). In addition, we have quantitated the amount of ceramide hydrolyzed by SIDT2 and analyzed by Michaelis-Menten equation (Fig. 5b and Supplementary Fig. 12 and the last paragraph, Page 13), the catalytic constant of SIDT2 in our detergent micelles condition is comparable to those for adiponectin receptor, alkaline ceramidase, and γ -secretase, another three kinds of intramembrane enzymes with very slow rates. Also, we found that SIDT2 cannot hydrolyze sphingomyelin, suggesting substrate specificity (Supplementary Fig. 13).

4. Is the cavity next to the Zn consistent with binding a hydrophobic molecule such as ceramide? Can the authors speculate on the enzymatic mechanism of ceramide hydrolysis based on the structure of the active site?

We thank this reviewer for the insightful comments. Structural comparison of SIDT2 with adiponectin receptor ADIPOR2 and alkaline ceramidase ACER3 reveals that they share a very similar Zn²⁺-dependent catalytic core (Fig. 4). Combined with the structural analysis of ADIPOR2 and ACER3 in complex with the oleic acid and monoolein, the cavity next to the Zn is consistent with binding a hydrophobic molecule such as ceramide. Actually, Sébastien

Granier group has already proposed a general acid-base catalytic mechanism (*Nat Commun.* 2018; 9:5437). In this mechanism, the zinc ion activates a water molecule for nucleophilic attack of the amide carbon in which D92 acts as a proton acceptor/donor. As the H3-D-S motif is highly conserved among these ceramidases, suggesting that they share a same enzymatic mechanism.

5. *Fig. 5a: where is the ceramide 1+ peak?*

Point well taken. The signal for ceramide 1+ peak was weak in the last time, then we provided the 2+ peak in the previous version. We have already reproduced this assay several times with increasing the substrate concentration from 20 μM to 40 μM , and the signal for the 1+ peak can be strongly detected. A representative mass spectrum for the extracted ion peaks of ceramide (d18:1/18:0) was provided as **Supplementary Fig. 13**.

6. *Supplementary Fig. 1a: The purified SIDT2 does not appear to be very pure as there are several significant bands above and below SIDT2. Could these contaminants play a role in the hydrolysis experiments? Mutagenesis of key catalytic residues would clarify this.*

We appreciate this excellent suggestion. To consolidate the ceramidase activity of SIDT2, we have generated a H796A-H800A mutant that each residue is crucial for binding the putative zinc ion. Expectedly, the mutant was devoid of the ceramidase activity. In addition, a “reaction buffer plus ceramide (d18:1/18:0)” sample and a “SIDT2 alone” sample act as controls (**Fig. 5a and Supplementary Fig. 11**).

7. *How do the authors explain the additional density appearing in the SIDT2 structure with added RNA? Presumably the RNA addition is the only difference with the other samples. Could RNA binding be stabilizing the C-terminal region?*

We thank this reviewer for the insightful comments. As the resolution is extremely low at this region, we cannot reliably distinguish whether it belongs to the protein or the RNA. Based on the predicted structure of SIDT2, a portion of the CTD was predicted to be located in the dimer interface (Supplementary Fig. 8), suggesting that the additional area of density is more likely a portion of the CTD. It has been reported that an arginine-rich motif in the CTD is capable of binding nucleic acids (*Autophagy.* 2020; 16: 1974–1988.). As the CTD is invisible in our cryo-EM structure, reflecting its intrinsic flexibility, the binding of RNA may stabilize this region. We have added it in the Discussion section.

Minor points

1. *All occurrences of 'systematic RNAi', presumably should be 'systemic RNAi'.*

Thanks. We have revised it.

2. *L151 – 152: 11 b-strands instead of 11 b-sheets.*

Thanks. We have revised it.

3. *L156: BRD1 and BRD2.*

Thanks. We have revised it.

4. *L152 and L158: What is meant by lamellar?*

Thanks. We have removed it and rewritten these sentences (The 2nd paragraph, Page 8).

5. *Fig. 3b and 3c: the labels are occluding the structural details; they should be moved away from structure and/or use thin arrows from label to residue.*

Point taken! We have revised it.

Reviewer #3:

Major points

1. *The significance of this study is weak. The authors could not show the structural basis for nucleic acids transport by SIDT2. The relationship among the SIDT2 structure, nucleic acids transport activity, and ceramidase activity needs to be investigated.*

We thank this reviewer for the critical comments. Please also see the first point in the “A brief summary of major revisions” section. So far, it is unclear that whether the SID1 family proteins function as a channel or a transporter. According to the literature, SIDT2 seems to be a transporter. Our structure also indicates that SIDT2 may act as a transporter but not a channel. We have confirmed that our SIDT2 protein is also required for transporting dsRNA (Supplementary Fig. 1). According to the literatures, besides nucleic acid transport, SIDT2 also has other diverse functions, including autophagy, glucose and lipid metabolism. Our data reveal that the SID1 family is a novel class of lipid hydrolases, advancing the understanding of the novel structure-function relationships, especially their functional roles in lipid metabolism. Taken together, our data marks an important step towards elucidating the molecular mechanisms of this important SID1 family proteins with varied functions. For clarity, we have revised these issues throughout in the Title, Abstract, Introduction, and Discussion (highlighted paragraphs).

2. *To clarify the relationship between structure and activity, whether or not the SIDT2 protein used in this study possesses an RNA transport activity needs to be shown. For example, does the SIDT2 construct enhance the import of extracellular RNA into the cells when the protein is overexpressed in cells?*

We have performed cell-based dsRNA intracellular localization assay and confirmed that our SIDT2 protein is also required for transporting dsRNA (Supplementary Fig. 1), suggesting that SIDT2 is not only a transporter but also an intramembrane enzyme.

3. *The possibility that the ceramidase activity is due to contaminated proteins is not excluded. To show that SIDT2 indeed exhibits ceramidase activity, the activity of SIDT2 mutant(s) needs to be investigated. For example, mutation(s) on the zinc-binding site (H3-D-S) may interfere with the ceramidase activity.*

We appreciate this excellent suggestion. To consolidate the ceramidase activity of SIDT2, we have generated a H796A-H800A mutant that each residue is crucial for binding the putative zinc ion. Expectedly, the mutant was

devoid of the ceramidase activity. In addition, a “reaction buffer plus ceramide (d18:1/18:0)” sample and a “SIDT2 alone” sample act as controls (Fig. 5a and Supplementary Fig. 11).

4. *It is unclear whether ceramide is a physiological and main substrate of SIDT2. Does SIDT2 hydrolyze ceramide in cells? Is the ceramidase activity of SIDT2 stronger or weaker than other known ceramidases? The ceramidase activity of SIDT2 can be predicted easily, because SIDT2 has a homology to other ceramidases (Pei et al. Biology Direct 2011, 6:37). Hydrolase activity for other lipids should also be investigated.*

We thank this reviewer for the insightful comments. We have quantitated the amount of ceramide hydrolyzed by SIDT2 and analyzed by Michaelis-Menten equation (Fig. 5b and Supplementary Fig. 12 and the last paragraph, Page 13), the catalytic constant of SIDT2 in our detergent micelles condition is comparable to those for adiponectin receptor, alkaline ceramidase, and γ -secretase, another three kinds of intramembrane enzymes with very slow rates. Also, we found that SIDT2 cannot hydrolyze sphingomyelin, reflecting its substrate specificity (Supplementary Fig. 13).

Actually, it has been reported that SIDT2 plays important roles in lipid metabolism, SIDT2-deficient mice exhibited an increase in serum triglycerides and free fatty acids (*Biochem Biophys Res Commun.* 2016; 476:326-332), a remarkable accumulation of lipid droplets in the liver (*J Lipid Res.* 2018; 59:404-415.), and changes in lysosomal membrane permeabilization and lipid metabolism (*Exp Ther Med.* 2018; 16:246-252). Genome-wide association studies revealed that SIDT2 was associated with high-density lipoprotein cholesterol levels and premature coronary artery disease (*Arterioscler Thromb Vasc Biol.* 2021; 41:2494-2508.) (The last paragraph, Page 4). Taken together, the ceramide hydrolysis by SIDT2 may be biologically relevant.

5. *Pratt et al. reported that the extracellular domain of SIDT1 forms a stable tetramer (Pratt et al. PLoS ONE 2012 e33607). Please explain the discrepancy between the current study and this paper.*

We thank this reviewer for the insightful comments. Actually, the conditions in our study are entirely different with that of the previous study. Our protein is intact and our structure reveals that the two TMDs also play important roles in forming a dimer, which may be the main reason for this discrepancy. In addition, we did not find any other oligomeric state in the purification step (Supplementary Fig. 2) and 2D or 3D classification step. We have added it in the Discussion (3rd paragraph, Page 15).

Minor point

1. *Fig. 5c. The illustration is wrong. Ceramide, sphingosine and fatty acids should not penetrate a lipid bilayer, but should localize to the inner or outer layer.*

Thanks. We have revised it.

REVIEWERS' COMMENTS

Reviewer #1 (Remarks to the Author):

The authors have for the most part addressed all of my comments and in my opinion, the manuscript is suitable for publication. However, I would make a few suggestions to refine the manuscript.

1. When referencing the comparison of catalytic rates of adiponectin receptors and alkaline receptors, it would be important to reference the adiponectin receptor manuscript in this sentence, and also make clear that the estimated k_{cat} for alkaline receptor is purely an estimate based on the amount of protein in microsomes and has not been directly determined.
2. SIDT2 is not a novel class of enzyme, as it shares the same active site residues and architecture as alkaline ceramidases and adiponectin receptors from the CREST family of enzymes.
3. It seems premature to mention the k_{cat} rate comparison with alkaline ceramidases in the abstract, as that value is only an estimate.
4. It is unclear why other sphingolipids (e.g. glucosylceramide) are not available for purchase, as they are available from Avanti polar lipids, which both ceramide and sphingomyelin were purchased from.

Reviewer #2 (Remarks to the Author):

Most of my points from the first review were addressed adequately.

However, given that the authors' own new experiments confirm the role of SIDT2 as an RNA transporter, with the structure in hand, the authors are in a unique position to be the first to make structure-guided mutations to the clusters of basic residues (not 1 by 1) in the BRDs shown in Supp. Fig. 7. This would significantly strengthen the functional role of this protein in RNA transport.

Minor points:

L190 There is no supplementary Fig. 4d

L190 “which was also identified”

There are still several grammatical errors that should be corrected.

Reviewer #3 (Remarks to the Author):

The authors addressed all my points. I have no other comments.

Response to Reviewers' Comments:

Reviewer #1:

The authors have for the most part addressed all of my comments and in my opinion, the manuscript is suitable for publication. However, I would make a few suggestions to refine the manuscript.

We thank this Reviewer for his/her positive feedback.

1. When referencing the comparison of catalytic rates of adiponectin receptors and alkaline receptors, it would be important to reference the adiponectin receptor manuscript in this sentence, and also make clear that the estimated k_{cat} for alkaline receptor is purely an estimate based on the amount of protein in microsomes and has not been directly determined.

Point taken. We have revised them according to the reviewer's suggestions (1st paragraph, Page 14).

2. SIDT2 is not a novel class of enzyme, as it shares the same active site residues and architecture as alkaline ceramidases and adiponectin receptors from the CREST family of enzymes.

We thank this reviewer for this insightful comment. We have removed this statement throughout the manuscript.

3. It seems premature to mention the k_{cat} rate comparison with alkaline ceramidases in the abstract, as that value is only an estimate.

Point taken. We have removed it from the abstract.

4. It is unclear why other sphingolipids (e.g. glucosylceramide) are not available for purchase, as they are available from Avanti polar lipids, which both ceramide and sphingomyelin were purchased from.

Actually, the website of MERCK, the exclusive partner of Avanti® Polar Lipids for all countries except the United States, clearly indicated that another several lipids are not currently available in China. We tried to contact with a lot of local agents, unfortunately, no supply was found. We are sorry about this. The finding of SIDT2 can't hydrolyze the sphingomyelin (d18:1/18:0) indicates that the larger polar headgroups may cause steric hindrance.

Reviewer #2:

Most of my points from the first review were addressed adequately.

We thank this Reviewer for his/her positive feedback.

However, given that the authors' own new experiments confirm the role of SIDT2 as an RNA transporter, with the structure in hand, the authors are in a unique position to be the first to make structure-guided mutations to the clusters of basic residues (not 1 by 1) in the BRDs shown in Supp. Fig. 7. This would significantly strengthen the functional role of this protein in RNA transport.

In this study, we mainly focused on revealing its lipid hydrolytic properties. We appreciate this suggestion and respectfully think it's more effective and targeted to mutate the key contact basic residues based on the structure of SID1-RNA complex in the future.

Minor points:

L190 There is no supplementary Fig. 4d

Point taken. We have revised it.

L190 "which was also identified"

Point taken. We have revised it.

There are still several grammatical errors that should be corrected.

Point taken. We have carefully read the manuscript and revised those grammatical errors.

Reviewer #3:

The authors addressed all my points. I have no other comments.

Thanks.